# Incidence of Osteoporosis in Primary Care Patients with Atrial Fibrillation Receiving Different Oral Anticoagulants

**DOI:** 10.3390/jcm11216438

**Published:** 2022-10-30

**Authors:** Woldesellassie M. Bezabhe, Jan Radford, Barbara C. Wimmer, Mohammed S. Salahudeen, Ivan Bindoff, Gregory M. Peterson

**Affiliations:** 1School of Pharmacy and Pharmacology, University of Tasmania, Private Bag 26, Hobart, TAS 7001, Australia; 2Launceston Clinical School, Tasmanian School of Medicine, University of Tasmania, 41 Frankland St., Launceston, TAS 7250, Australia

**Keywords:** osteoporosis, oral anticoagulants, warfarin, dabigatran, rivaroxaban, apixaban, atrial fibrillation, primary care

## Abstract

Background: Studies investigating the association between the use of oral anticoagulants (OACs) and osteoporosis are limited. We aimed to determine the risk of osteoporosis in patients with atrial fibrillation (AF) and receiving different OACs. Methods: We performed a population-based cohort study using a nationwide primary care dataset, MedicineInsight. Patients aged between 18 and 111 years with AF and newly recorded OAC prescriptions between 1 January 2013 and 31 December 2017 were included and followed until 31 December 2018. We applied propensity score matching to control for patients’ baseline characteristic differences before calculating adjusted hazard ratios (aHRs) for a new diagnosis of osteoporosis, using Cox proportional hazard models. Results: A total of 18,454 patients (1714 prescribed dabigatran, 5871 rivaroxaban, 5248 apixaban and 5621 warfarin) were included. Of these, 39.5% were females, and the overall mean age (standard deviation [SD] was 73.2(10.3) years. Over a mean follow-up of 841 days, 1627 patients (1028 receiving direct-acting oral anticoagulants (DOACs) and 599 warfarin) had a newly recorded diagnosis of osteoporosis. The weighted incidence rates (95% confidence interval; CI) per 100 person-years of treatment were 5.0 (4.7–5.2) for warfarin, 4.3 (3.8–4.8) for dabigatran, 3.6 (3.3–3.8) for rivaroxaban, and 4.4 (4.0–4.7) for apixaban. Overall, DOAC use was associated with a significantly lower risk of a new diagnosis of osteoporosis than warfarin use (aHR, 0.79, 95% confidence interval (CI) 0.74–0.85; *p* < 0.001). Use of each individual DOAC was associated with a significantly lower risk of osteoporosis compared with warfarin (aHRs, 0.75, 95% CI 0.69–0.82 for rivaroxaban; 0.78, 95% CI 0.71–0.86 for apixaban; 0.88, 95% CI 0.77–0.99 for dabigatran). Conclusion: Compared with warfarin, the use of DOACs was associated with a significantly lower risk of developing osteoporosis in patients with AF. This association remained significant when individual DOACs were compared with warfarin.

## 1. Introduction

Osteoporosis is the most common chronic metabolic bone disease, characterised by bone microarchitecture, density, and strength impairments. It increases the risk of bone fracture and is a common contributor to death and disability in the older population [1]. With the aging population worldwide, the number of people with osteoporosis is increasing, as are people with atrial fibrillation (AF) [1,2].

Limited previous studies indicate that the use of warfarin, a vitamin K antagonist and the cornerstone of stroke prevention in patients with AF for decades, increases the risk of osteoporosis and bone fractures [3]. Vitamin K deficiency induced by warfarin can cause osteoporosis and osteoporotic fractures by altering bone metabolism [4]. The warfarin alternatives, direct-acting oral anticoagulants (DOACs), are equal or superior in efficacy and safety in preventing stroke in patients with AF [5], and their use rates now exceed warfarin [6].

In our recent analysis, we found that the risk of being diagnosed with new-onset osteoporosis increased in patients with AF prescribed either DOACs or warfarin compared with those not prescribed an OAC [7]. Few studies have examined the risk of osteoporosis associated with the use of DOACs, compared with warfarin [8,9]. Huang et al. reported a significantly lower risk of osteoporosis in patients receiving DOACs compared with warfarin (adjusted hazard ratio (aHR), 0.68, 95% confidence interval (CI), 0.55–0.83) [8]. However, the diagnostic codes used to define osteoporosis in their dataset were not benchmarked against patients’ osteoporosis status, and misclassifications could have resulted in misleading associations. With MedicineInsight, the Australian dataset for this study, diagnostic coded terms for osteoporosis were clinically validated and had excellent accuracy [10]. We aimed to extend our previous work [7] by evaluating the risks of osteoporosis in patients with AF and receiving a DOAC compared to either another DOAC or warfarin using the validated MedicineInsight dataset. 

## 2. Methods

We performed a retrospective cohort study using MedicineInsight, a large-scale primary care dataset of longitudinal de-identified electronic health records (EHRs) in Australia. The MedicineInsight program collates routinely collected EHR data from clinical information systems from consenting general practices, which have agreed to provide data on an ongoing basis. As of March 2020, the EHRs of 732 general practices, which provided clinical care for over 3 million patients across Australia, were extracted into the MedicineInsight dataset [11]. The dataset consists of sociodemographic information, diagnoses, prescriptions, pathology tests and observations recorded by general practitioners while providing clinical care [11]. By comparing with Medicare Benefits Schedule data, it was determined that MedicineInsight represents the Australian population in terms of age and sex [12]. 

Comorbidities in this study were based on condition flags provided by MedicineInsight and developed using coding algorithms from three EHR fields: diagnosis, the reason for the visit, and the reason for a prescription [11]. The coded terms for the conditions included in this study are shown in our previous article [7]. MedicineInsight algorithms demonstrate excellent accuracy in identifying patients with osteoporosis [10]. Details about the MedicineInsight dataset are available elsewhere [6,12,13]. 

The study population included patients aged between 18 and 111 years, with a recorded diagnosis of AF and no missing information on sex. The patients had to have a recorded prescription of an oral anticoagulant (OAC) between 1 January 2013 and 31 December 2017 (screening period) but no recorded OAC prescription within the two years prior to this screening period. All four OACs (warfarin, dabigatran, rivaroxaban, and apixaban) were available in Australia during the screening period. We excluded patients who had less than three recorded visits to their general practice within two years (within the index year and the year before). These patients were less likely to be active patients managed by the practice contributing data. Patients who had a recorded diagnosis of osteoporosis before the index date (the first date that an OAC was prescribed in the screening period) were also excluded, as were patients who received OAC treatment for less than six months. At least 6 months of follow-up was required to provide time for the development of osteoporosis.

A newly recorded diagnosis of osteoporosis after the index date was defined as the primary outcome. We followed patients until the diagnosis of osteoporosis, OAC discontinuation, a switch from the index OAC, or the end of the study period (31 December 2018), whichever occurred first. The OAC discontinuation date was defined as the date the supply, obtained during follow-up using the last recorded OAC prescription in the study period, potentially ended. In Australia, the supply obtained from an OAC prescription normally lasts 180 days. The switch date was the first date an OAC other than the index OAC was prescribed during follow-up to a patient who had been receiving the index OAC. We did not follow patients once they were switched to a different OAC; this was to compare the association between individual OAC use and a subsequent diagnosis of osteoporosis. 

Baseline patient characteristics were compared using descriptive statistics. The treatment effect was estimated using a propensity score with inverse probability of treatment weighting (IPTW). Absolute standardised differences of covariates ≤0.10 were considered negligible [14]. The weighted incidence rates were reported as events per 100 person-years. Baseline demographics, conditions, and medications were included in the propensity models (Table 1). The index year was included in the model to balance the length of follow-up among treatment groups. The hazard of diagnosis with osteoporosis for patients receiving a DOAC was compared to those with warfarin using a Cox proportional hazards model. We included all covariates listed in Table 1 to adjust the multivariable Cox proportional hazards models. 

We performed subanalyses based on sex, age, and duration of OAC use. Furthermore, five sensitivity analyses were also conducted to validate the main analysis. The first was performed without adjusting baseline characteristic differences with propensity score matching. The second sensitivity analysis was performed after matching with greedy nearest neighbour propensity score matching. We used a calliper width of 0.2 on the logit of propensity score [15]. Warfarin might be a preferred OAC in patients with chronic liver disease [16], advanced-stage chronic kidney disease (stage 4 and 5, coded terms provided in our previous publication [7]), or rheumatic heart disease; that could have introduced some confounding by indication [17], and so the third sensitivity analysis was performed by excluding patients with these conditions to minimise this bias. The fourth sensitivity analysis was performed comparing groups based on their index OAC but without censoring when switching from their index OAC. The last sensitivity analysis was performed by excluding patients with at least one recorded prescription of systemic corticosteroids, antiepileptics, or proton pump inhibitors, which may increase the risk of osteoporosis [7]. 

A two-sided *p* value ≤ 0.05 was considered statistically significant. Data management and analyses were performed using SAS software (SAS version 9.4, SAS Institute Inc., Cary, NC, USA). The reporting of the study follows the EQUATOR guidelines [18]. 

## 3. Results

A total of 18,454 patients with AF (39.5% were females;) and mean age (standard deviation [SD] of 73.2 [10.3] years) and who had a newly recorded OAC prescription between 1 January 2013 and 31 December 2017 were included. Of these, 1714 were prescribed dabigatran, 5871 rivaroxaban, 5248 apixaban, and 5621 warfarin (Figure 1). The mean duration of follow-up (SD) was 841 (485) days and ranged from 777 (392) days for apixaban to 878 (511) days for dabigatran. Before adjustment with IPTW, one-third of the baseline variables had standardised mean differences greater than 0.1. After adjustment with IPTW, all baseline characteristics had weighted standardised mean differences less than 0.1 (Table 1). 

A total of 1627 patients (1028 receiving DOACs and 599 warfarin) had a newly recorded diagnosis of osteoporosis following treatment with OACs. The weighted incidence rates (95% CIs) per 100 person-years were 5.0 (4.7–5.2) for those prescribed warfarin, 4.3 (3.8–4.2) for dabigatran, 3.6 (3.3–3.8) for rivaroxaban and 4.4 (4.0–4.7) for apixaban (Table 2). The weighted median (interquartile range) days from the first date of an OAC prescription to the first date of an osteoporosis diagnosis were 523 (315–858) for those prescribed warfarin, 641 (379–1057) dabigatran, 567 (334–951) rivaroxaban and 546 (332–873) apixaban.

Overall, DOAC use was associated with a significantly lower risk of a new diagnosis of osteoporosis than warfarin (aHR, 0.79, 95% CI 0.74–0.85; *p* < 0.001). Comparing individual DOACs with warfarin, we found that all three DOACs were associated with a lower risk of osteoporosis (aHRs, 0.75, 95% CI 0.69–0.82 for rivaroxaban; 0.78, 95% CI 0.71–0.86 for apixaban; 0.88, 95% CI 0.77–0.99 for dabigatran). In our subanalyses comparing the risk of newly diagnosed osteoporosis between individual DOACs, we found that osteoporosis was significantly lower in patients treated with rivaroxaban than dabigatran (aHR, 0.83, 95% CI 0.72–0.95; *p* < 0.01). However, there were no significant differences in the risk of osteoporosis in patients receiving apixaban compared with those receiving either dabigatran or rivaroxaban (Table 3). 

After stratification by sex, the risk of osteoporosis remained significantly lower in both males (aHR, 0.79, 95% CI, 0.71–0.89; *p* < 0.001) and females (aHR, 0.80, 95% CI, 0.73–0.87; *p* < 0.001) receiving a DOAC compared with warfarin. The risk of osteoporosis was significantly lower in patients aged ≥60 years who had received a DOAC (aHR, 0.78, 95% CI, 0.73–0.84; *p* < 0.001) than warfarin. However, with a smaller sample, this risk was not significantly different in patients aged <60 years receiving a DOAC compared with warfarin (aHR, 0.78, 95% CI, 0.39–1.56; *p* = 0.463). With subanalysis based on the duration of OAC treatment, the risk of osteoporosis was significantly lower for DOACs than warfarin only in those patients who had received OAC treatment for longer than one and half years (548 days) (aHR, 0.78, 95% CI, 0.71–0.87; *p* < 0.001). It was not significantly different between these two groups when patients had received the OAC treatment for a shorter period (aHR, 0.97, 95% CI, 0.88–1.07; *p* = 0.51) (Table 4). 

The five sensitivity analyses performed (including all eligible patients without adjusting their baseline characteristic differences; using adjusted baseline characteristic differences based on nearest-neighbour propensity score matching; excluding patients who had chronic liver disease, rheumatic heart disease, or chronic kidney disease stage 4 or 5; without censoring patients for switching their index OAC; and excluding patients with at least one recorded prescription of systemic corticosteroids, antiepileptics, or proton pump inhibitors) showed findings similar to the primary analysis (Table 5). 

## 4. Discussion

Using Australian primary care data, we found that DOAC use, compared with warfarin, was associated with a lower risk of osteoporosis in patients with AF. With subanalysis by the duration of OAC treatment, we demonstrated that the difference in risk of osteoporosis tended to be greater with a longer duration of treatment. The risks of osteoporosis remained significantly lower comparing each DOAC (dabigatran, rivaroxaban, apixaban) with warfarin. This study differed from our previous one [7], which demonstrated an increase in risk of osteoporosis in patients prescribed OACs compared with those not prescribed an OAC, by focusing on comparing the risk of osteoporosis associated with the use of DOACs, including individual agents, relative to warfarin. 

Our findings are consistent with the study by Huang et al. [8] that reported a lower risk of osteoporosis in patients who received a DOAC than warfarin using ‘Taiwan’s National Insurance Research Database. The lower risk of osteoporosis in patients who received a DOAC than warfarin was stronger when OAC treatment duration was longer or patients were aged 60 years or over, which agrees with our subanalyses. In contrast with our study, Huang et al. [8] did not find a significant difference in risk of osteoporosis in males who received a DOAC compared with warfarin. Two possible explanations for the observed difference could be our longer duration of follow-up (mean follow-up 841 (485) days compared with median follow-up 2.1 years for Huang et al. [8]) and slightly older cohort (73.2 (10.3) years for this study vs. 71.0 (11.4) years (DOAC cohort) and 70.8 (11.9) years (warfarin cohort) for Huang et al. [8]). A study by Lau et al. using the Hong Kong Hospital Authority database found a consistently lower risk of osteoporotic fractures in females and males with AF (mean age 74.4 (10.8) years) treated with DOACs compared with warfarin [3]. 

Each of the three DOACs (dabigatran, rivaroxaban, and apixaban) was associated with a significantly lower risk of osteoporosis when compared with warfarin. This was consistent with Lau et al. [3], who reported that the use of each of the three same DOACs was associated with a lower risk for fracture than the use of warfarin. However, the Huang et al. [8] study did not find a lower risk of osteoporosis in patients who had received dabigatran compared with those with warfarin. Misclassification of patients by Huang et al. might be the reason for osteoporosis risk in the dabigatran group not differing from warfarin. Although their study involved more patients than our study, the event rate they reported was low. The incidence rate of newly recorded osteoporosis in the dabigatran group was less than half of ours (44 vs. 18.5 per 1000 person-years). This could have been due to misclassification of osteoporosis outcomes in ‘Taiwan’s National Insurance Research Database [8], in which osteoporosis diagnostic codes were not validated. In contrast, the diagnostic coded terms for osteoporosis in our dataset, MedicineInsight, were validated by directly evaluating patients and had excellent accuracy [10]. 

## 5. Strengths and Limitations 

This is the first study to evaluate the risk of osteoporosis in patients with AF and treated with individual OACs, using an Australian national primary care dataset. Compared with the Medicare Benefits Scheme data, the MedicineInsight data of regular patients represent the Australian population’s sex and age (9). We used IPTW to adjust baseline patient characteristic differences. All the four OACs were available in Australia during the screening period, 2013–2017. However, the study has several limitations. Baseline patient characteristic differences, such as body mass index, bone mineral density, vitamin D prescribing, total hysterectomies or bilateral oophorectomies in female patients, alcohol consumption and smoking status, were not assessed; thus, cohort matching was not fully complete. However, the cohorts were adjusted by including several measured baseline characteristics in the IPTW models. Some of these partly count for the unmeasured variables not assessed. For instance, chronic obstructive pulmonary disease may partially account for smoking status. Besides, the variables not assessed mentioned above do not typically determine the choice of OAC and are not likely to have caused confounding by indication. The sensitivity analysis results, after excluding patients who could be preferably prescribed warfarin based on their recorded baseline characteristics (chronic liver disease [16], rheumatic heart disease, chronic kidney disease stage 4 or 5), were similar to the main analysis. 

The use of a DOAC is associated with a significantly lower risk of bleeding than warfarin [5]. Patients with a high fall risk at baseline might have been preferably prescribed a DOAC than warfarin. Falls were not flagged in our MedicineInsight extracted data and not included in our matching, which might have introduced bias. We assumed that patients who had recorded OAC prescriptions were taking their medication as directed during follow-up. We used prescribed data and had no means to confirm whether the prescriptions were filled or the patients took the OACs. 

## 6. Conclusions

In this study using a propensity score-matched Australian primary care cohort of patients with AF, DOAC use was associated with a significantly lower risk of osteoporosis than warfarin, which remained consistent when stratified by sex and individual DOACs. The association of lower osteoporosis risk with DOACs, compared with warfarin, tended to be more pronounced with longer OAC treatment duration and in patients aged 60 years or over. 

## Figures and Tables

**Figure 1 jcm-11-06438-f001:**
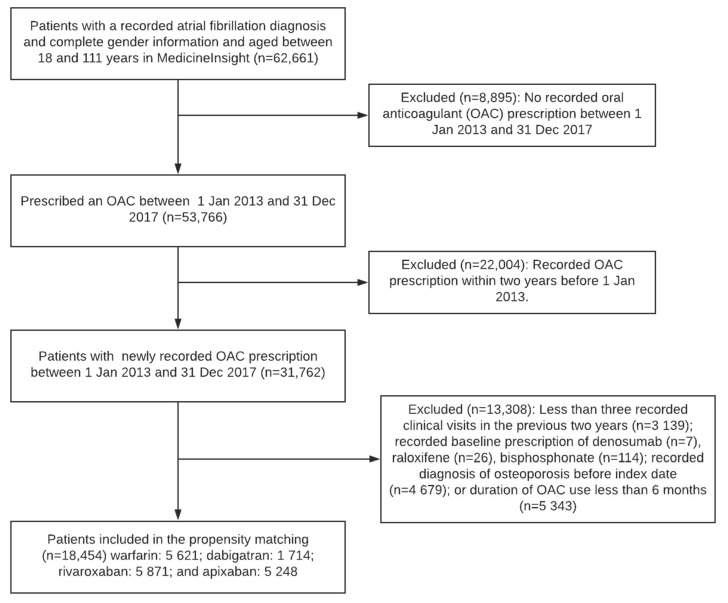
Selection of cohort.

**Table 1 jcm-11-06438-t001:** Baseline characteristics.

Characteristics	DOACs			Warfarin	Standardised Mean Difference	
	Dabigatran	Rivaroxaban	Apixaban		Before IPTW	After IPTW
Patients, *n*	1714	5871	5248	5621		
Age, yrs; mean (SD)	73.1(9.5)	71.8(10.0)	73.9(10.1)	74.1(10.7)	−0.122	0.004
Females, %	664 (38.7)	2181 (37.2)	2210 (42.1)	2238 (39.8)	0.009	−0.005
eGFR, mL/min/1.73 m^2^	73.8 (17.6)	74.8 (17.8)	72.6(18.3)	67.9 (21.3)	0.301	0.009
CHA_2_DS_2_-VASc; mean (SD)	3.7 (1.7)	3.4 (1.7)	3.7 (1.7)	4.0 (1.8)	−0.224	−0.005
Index year, *n* (%)					0.954	−0.041
2013	352 (20.5)	467 (8.0)	64 (1.2)	2300 (40.9)		
2014	359 (21.0)	1275 (21.7)	635 (12.1)	1239 (22.0)		
2015	245 (14.3)	1343 (22.9)	1172 (22.3)	897 (16.0)		
2016	310 (18.1)	1405 (23.9)	1568 (29.9)	660 (11.7)		
2017	448 (26.1)	1381 (23.5)	1809 (34.5)	525(9.3)		
Comorbidities						
Congestive heart failure, *n* (%)	408 (23.8)	1266 (21.6)	1254 (23.9)	1947 (34.6)	−0.264	−0.022
Hypertension, *n* (%)	1258 (73.4)	4122 (70.2)	3780 (72.0)	3990 (71.0)	0.009	0.003
Stroke or transient ischaemic attack, *n* (%)	344 (20.1)	928 (15.8)	928 (15.8)	1257(22.4)	−0.126	−0.004
Diabetes mellitus, *n* (%)	493 (28.8)	1597 (27.2)	1364 (26.0)	1713 (30.5)	−0.079	0.000
Peripheral vascular disease, *n* (%)	563 (32.9)	2084 (35.5)	1990 (37.9)	2345 (41.7)	−0.115	−0.002
Coronary heart disease, *n* (%)	421 (24.6)	1550 (26.4)	1507 (28.7)	1794(31.9)	−0.106	0.003
Anxiety, *n* (%)	261 (15.2)	948 (16.2)	832 (15.9)	848 (15.1)	0.023	−0.003
Depression, *n* (%)	393 (22.9)	1383 (23.6)	1304 (24.9)	1344 (23.9)	0.002	0.000
Dementia, *n* (%)	78 (4.6)	210 (3.6)	178 (3.4)	298 (5.3)	−0.081	−0.010
Arthritis, *n* (%)	1026 (59.9)	3439 (58.6)	3101 (59.1)	3101 (59.1)	−0.042	0.026
Asthma, *n* (%)	297 (17.3)	1048 (17.9)	948 (18.1)	996(17.7)	0.004	0.007
COPD, *n* (%)	269 (15.7)	883 (15.0)	796 (15.2)	1062 (18.9)	−0.099	0.007
Cancer, *n* (%)	746 (43.5)	2362 (40.2)	2230(42.5)	2230 (42.5)	0.001	0.016
RAAs inhibitors, *n* (%)	1232 (71.9)	4232 (72.1)	3777 (72.0)	4109 (73.1)	−0.024	0.005
Proton pump inhibitors, *n* (%)	925 (54.0)	2925 (49.8)	2735 (52.1)	3018 (53.7)	−0.048	0.012
Beta-blockers, *n* (%)	1201 (70.1)	4184 (71.3)	3859 (73.5)	4077 (72.5)	−0.011	0.009
Antiplatelet agents, *n* (%)	441 (25.7)	1849 (31.5)	1779 (33.9)	1628(29.0)	0.060	0.026
NSAIDs, *n* (%)	540 (31.5)	2090 (35.6)	1772 (33.8)	1314 (23.4)	0.243	0.046
Statins, *n* (%)	973 (59.7)	3355 (59.3)	3073 (60.7)	3186 (60.7)	−0.080	−0.002
SSRIs, *n* (%)	295 (18.1)	1072 (18.9)	984(19.5)	1067 (20.2)	−0.036	−0.007

COPD, chronic obstructive pulmonary disease; CHA_2_DS_2_-VASc (congestive heart failure (1 point), hypertension (1 point), age ≥ 75 years (2 points), diabetes mellitus (1 point), stroke/transient ischaemic attack (2 points), vascular disease (1 point), age 65–74 years (1 point) and sex female (1 point)); DOAC, directly-acting oral anticoagulant; eGFR, estimated glomerular filtration rate; IPTW, inverse probability of treatment weighting; NSAIDs, non-steroidal anti-inflammatory drugs; OAC, oral anticoagulant; SSRIs, selective serotonin reuptake inhibitors; RAAS, renin-angiotensin-aldosterone system.

**Table 2 jcm-11-06438-t002:** The incidence of newly-diagnosed osteoporosis in the study cohort.

Treatment	Total Patients, *n*	Median Follow-Up (IQR), Days	Osteoporosis, *n*	Person-Years	Crude Incidence per 100 Person-Years (95% CI)	Weighted Incidence per 100 Person-Years (after IPTW) (95% CI)
All patients						
Warfarin	5621	704 (371–1239)	599	12,240	4.9 (4.5–5.3)	5.0 (4.7–5.2)
DOAC	12,833	739 (472–1139)	1028	25,906	4.0 (3.7–4.2)	4.0 (3.8–4.2)
Dabigatran	1714	725 (462–1240)	164	3755	4.4 (4.4–5.1)	4.3 (3.8–4.8)
Rivaroxaban	5871	787 (484–1207)	439	12,487	3.5 (3.2–3.9)	3.6 (3.3–3.8)
Apixaban	5248	699 (462–1029)	425	9664	4.4 (4.0–4.8)	4.4 (4.0–4.7)
Total	18,454	728 (446–1162)	1627	38,146	4.3 (4.1–4.5)	4.4 (4.3–4.6)
Male						
Warfarin	3383	715 (375–1279)	229	7445	3.1 (2.7–3.5)	3.1 (2.9–3.4)
Dabigatran	1050	744 (465–1274)	60	2339	2.6 (2.0–3.3)	2.6 (2.2–3.1)
Rivaroxaban	3690	791 (484–1207)	173	7851	2.2 (1.9–2.6)	2.2 (1.9–2.5)
Apixaban	3038	714 (475–1050)	157	5683	2.8 (2.4–3.2)	2.8 (2.4–3.1)
Total	11,161	742 (451–1175)	619	23,318	2.7 (2.5–2.9)	2.7 (2.6–2.9)
Female						
Warfarin	2238	691 (366–1197)	370	4795	7.7 (7.0–8.5)	7.9 (7.4–8.4)
Dabigatran	664	700 (461–1213)	104	1416	7.3 (6.0–8.8)	7.0 (6.0–8.0)
Rivaroxaban	2181	777 (484–1204)	266	4636	5.7 (5.1–6.5)	5.9 (5.4–6.4)
Apixaban	2210	678 (447–1000)	268	3981	6.7 (5.9–7.6)	6.7 (6.1–7.4)
Total	7293	711 (434–1145)	1008	14,828	6.8 (6.4–7.2)	7.1 (6.8–7.4)

DOAC, direct-acting oral anticoagulant; IQR, interquartile range; IPTW, inverse probability of treatment weighting.

**Table 3 jcm-11-06438-t003:** Relative risk of osteoporosis after IPTW.

Treatment	All Patients	
	Hazard Ratio (95% CI)	*p* Value
DOAC vs. warfarin	0.79 (0.74–0.85)	<0.001
Dabigatran vs. warfarin	0.88 (0.77–0.99)	0.044
Rivaroxaban vs. warfarin	0.75 (0.69–0.82)	<0.001
Apixaban vs. warfarin	0.78 (0.71–0.86)	<0.001
DOAC vs. DOAC		
Rivaroxaban vs. dabigatran	0.83 (0.72–0.95)	0.008
Apixaban vs. dabigatran	0.87 (0.74–1.02)	0.079
Rivaroxaban vs. apixaban	0.96 (0.85–1.07)	0.44

CI, confidence interval; DOAC, direct-acting oral anticoagulant; IPTW, inverse probability of treatment weighting.

**Table 4 jcm-11-06438-t004:** Relative risk of osteoporosis in patients treated with DOACs versus warfarin based on sex, age, and treatment duration.

Variables	HR (95% CI)	*p* Value
Sex		
Male	0.79 (0.71–0.89)	<0.001
Female	0.80 (0.73–0.87)	<0.001
Age, years		
<60	0.78 (0.39–1.56)	0.463
≥60	0.78 (0.73–0.84)	<0.001
Duration of OAC use, days		
≤548 (one and half years)	0.97 (0.88–1.07)	0.510
>548	0.78 (0.71–0.87)	<0.001

CI, confidence interval; DOAC, direct-acting oral anticoagulant; HR, hazard ratio; OAC, oral anticoagulant. All baseline characteristics listed in Table 1 were included in the Cox proportional hazard models when calculating HRs.

**Table 5 jcm-11-06438-t005:** Comparison for the incidence of osteoporosis in patients treated with DOACs versus warfarin in sensitivity analyses.

Treatment	Sensitivity Analysis A *		Sensitivity Analysis B ^†^		Sensitivity Analysis C ^‡^		Sensitivity Analysis D ^§^	
	Adjusted HR (95% CI)	*p* Value	Adjusted HR (95% CI)	*p* Value	Adjusted HR (95% CI)	*p* Value	Adjusted HR (95% CI)	*p* Value
DOAC vs. warfarin								
DOAC overall	0.78 (0.70–0.88)	<0.001	0.79 (0.70–0.90)	0.006	0.78(0.73–0.84)	<0.001	0.93(0.86–0.99)	0.034
Warfarin	Reference		Reference		Reference		Reference	
Dabigatran vs. warfarin								
Dabigatran	0.89 (0.74–1.06)	0.188	0.90 (0.72–1.11)	0.325	0.86 (0.75–0.98)	0.021	0.99 (0.88–1.13)	0.928
Warfarin	Reference		Reference		Reference		Reference	
Rivaroxaban vs. warfarin								
Rivaroxaban	0.76 (0.66–0.87)	<0.001	0.78 (0.66–0.91)	0.002	0.75 (0.69–0.83)	<0.001	0.89 (0.82–0.98)	0.014
Warfarin	Reference				Reference		Reference	
Apixaban vs. warfarin								
Apixaban	0.84 (0.69–1.02)	0.081	0.74 (0.61–0.88)	0.001	0.77 (0.70–0.85)	<0.001	0.93 (0.85–1.03)	0.172
Warfarin	Reference		Reference		Reference		Reference	

CI, confidence interval; DOAC, direct-acting oral anticoagulant; HR, hazard ratio. * Sensitivity analysis A was performed by including all eligible patients without the inverse probability of treatment weighting. ^†^ Sensitivity analysis B was performed after matching with the greedy nearest neighbour propensity score. ^‡^ Sensitivity analysis C was conducted by excluding patients who had chronic liver disease, or rheumatic heart disease, or chronic kidney disease stage 4 or 5 before matching with inverse probability of treatment weighting. ^§^ Sensitivity analysis D was performed without censoring patients for switching from their index OAC but matching with inverse probability of treatment weighting. All baseline characteristics in Table 1 were included in the Cox proportional hazard models when calculating adjusted HRs.

## Data Availability

The authors’ license for using these data does not allow sharing raw data with third parties. However, other researchers are able to access these data in the same manner as the authors. Data access inquiries can be directed to NPS MedicineWise (https://medicineinsight@nps.org.au (accessed on 1 February 2022)).

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
