# Peer review of "Incidence of Osteoporosis in Primary Care Patients with Atrial Fibrillation Receiving Different Oral Anticoagulants"

_jcm, 2022, doi:10.3390/jcm11216438_

Round 1

Reviewer 1 Report

Thank you for your pioneer in this field of OAC and osteoporosis. I have read this with interest. There were several points that need to be revised in this manuscript . Although this revision needs further information and analysis but it is worth doing it.

1. The authors did not clarify how the diagnosis of osteoporosis was made with certainty other than the coding from  MedicineInsight algorithms .  Although, the authors stated that this algorithms demonstrated excellent accuracy in identifying patients with osteoporosis but did all the patients had bone mineral density(BMD)  confirmed or by other reliable means of diagnosis.

2. There was no baseline BMD or data confirming that the patients who were subsequently diagnosed with osteoporosis did not have this prior to or at the time of OAC exposure.

3.Degree of osteoporosis( either with T score or Z score) should also be mentioned in order to guide the chronicity and severity of the disease which might confound the results and showed some causal relationship .

4. Several major causes of osteoporosis are lacking in the cox proportional hazard  model  and subsequently propensity score matching as the authors stated that all variables in table 1 were used such as body mass index , history or current use of certain medications that might effect the bones (eg. steroids, Thiazolidinediones for DM treatment , calcium supplement),   thyroid function /status as in hyperthyroid and finally but very important is the vitamin D level. Even though  some of these parameters were mentioned in the Strengths and limitations section by the authors such as body mass index but it should not be too difficult to track the body weight which is also relevant for osteoporosis from the MedicineInsight algorithms . 

5. Whether any of these female patients had surgery of ovary in the past was also not mentioned.

6. Spatial relationship of the time of OAC exposure and time at first diagnosis should also be identified since a few months of OAC exposure and then the  diagnosis of osteoporosis would not be  reasonable.

7. How do the authors confirm adherence and how often the INR been measured in warfarin arm. Also , the  dose of OAC  should also be clarified and any dosed adjustment during the period of treatment?

8. Means of presentation such as graph or bars can be implement other that tables.

    I wish you could add more data on what I have requested.

Regards

Author Response

Reviwer 1.

Thank you for your pioneer in this field of OAC and osteoporosis. I have read this with interest. There were several points that need to be revised in this manuscript . Although this revision needs further information and analysis but it is worth doing it.

Point 1. The authors did not clarify how the diagnosis of osteoporosis was made with certainty other than the coding from  MedicineInsight algorithms. Although, the authors stated that this algorithms demonstrated excellent accuracy in identifying patients with osteoporosis but did all the patients had bone mineral density(BMD)  confirmed or by other reliable means of diagnosis.

Response 1. It is true that coded and non-coded terms were used for identifying patients with a recorded osteoporosis diagnosis in MedicineInsight. However, the diagnoses were made based on the Royal Australian College of General Practitioners and Osteoporosis Australia guidelines, in which confirming patients' bone mineral density (BMD) using Dual-energy X-ray absorptiometry (DXA) is essential [1]. This is included in Methods section, page 6, lines 122-124.

Point 2. There was no baseline BMD or data confirming that the patients who were subsequently diagnosed with osteoporosis did not have this prior to or at the time of OAC exposure.

Response 2: As a retrospective study, important baseline variables that are supposed to be collected for a typical osteoporosis study in prospective design, such as BMD, were not recorded for all patients. We relied on excluding patients diagnosed with osteoporosis before OAC exposure.

Point 3. Degree of osteoporosis( either with T score or Z score) should also be mentioned in order to guide the chronicity and severity of the disease which might confound the results and showed some causal relationship .

Response 3: The Royal Australian College of General Practitioners and Osteoporosis Australia guidelines  measure the degree of osteoporosis using a T score. However, the MedicineInsight data did not grade osteoporosis based on the T score.

 Point 4. Several major causes of osteoporosis are lacking in the cox proportional hazard  model  and subsequently propensity score matching as the authors stated that all variables in table 1 were used such as body mass index , history or current use of certain medications that might effect the bones (eg. steroids, Thiazolidinediones for DM treatment , calcium supplement),   thyroid function /status as in hyperthyroid and finally but very important is the vitamin D level. Even though  some of these parameters were mentioned in the Strengths and limitations section by the authors such as body mass index but it should not be too difficult to track the body weight which is also relevant for osteoporosis from the MedicineInsight algorithms . 

Response 4: Vitamin D is available over the counter. The recorded vitamin D prescription did not reflect the actual use. Similarly, only a few patients had recorded body mass indexes. Therefore, we prefered not to include them in our analysis and mentioned the limitations.

Point 5. Whether any of these female patients had surgery of ovary in the past was also not mentioned.

Response 5: Female patients with total hysterectomies or bilateral oopherectomies were not flagged in the MedicineInsight dataset; hence the effect of surgical removal of ovaries was not adjusted in the models. This is now mentioned as a limitation in the Strengths and limitations section, page 11, line 239.

Point 6. Spatial relationship of the time of OAC exposure and time at first diagnosis should also be identified since a few months of OAC exposure and then the  diagnosis of osteoporosis would not be  reasonable.

Response 6. The minimum follow-up time was 6 months, similar to our previous study [2]. This means we already excluded patients who were diagnosed with new-onset osteoporosis within the first 6 months of OAC exposure. This is added in the Methods section: “At least 6 months of follow-up was required as a prior study in patients prescribed warfarin did not find osteoporosis within a year (6) and assumed osteoporosis occurred prior to this timeframe.”

Point 7. How do the authors confirm adherence and how often the INR been measured in warfarin arm. Also , the  dose of OAC  should also be clarified and any dosed adjustment during the period of treatment?

Response 7. We already mentioned not measuring adherence as a limitation in the Strengths and limitations section as "We assumed that patients who had recorded OAC prescriptions were taking their medication as directed during follow-up. We used prescribed data and had no means to confirm whether the prescriptions were filled or the patients took the OACs.".

We agree that doses of OAC are adjusted in patients with compromised drug clearance (kidney disease), as is clinically appropriate. For instance, the reduced dose of apixaban, 2.5 mg twice a day, is prescribed if at least two of the three criteria (weight <60 kg, age>80 years, serum creatinine >133 micromol/L) are fulfilled. However, this dose adjustment is to intentionally match the lower drug clearance in elderly people or with advanced chronic kidney disease. At steady-state, the plasma concentration of the apixaban is expected to be the same regardless of the apixaban dose patients are taking. That is, the concentration of the apixaban that reaches the 'site of action' to cause the therapeutic/adverse effect (osteoporosis) in patients with compromised apixaban clearance (receiving 2.5 mg twice a day) will be similar to those receiving the normal apixaban dose (5 mg twice a day). As far as the dose adjustment is appropriate, taking a high or low dose of DOAC does not indicate a different exposure to the drug or risk for osteoporosis. However, we understand that some patients might have received an off-label low dose of DOAC. Their risk for osteoporosis might differ from those taking it based on clinical guideline recommendations.

Point 8. Means of presentation such as graph or bars can be implement other that tables.

Response 8. It is unclear which table to present in the graph or bars.

Reviewer 2 Report

The authors present an interesting review on a topic that, is relatively new and poorly explored by now, albeit not exactly central in cardiovascaular medicine practice

 The description of the method is clear and concise, with sufficient details to understand the methods and the rationale.

 The language is correct, the paper is well written, every step needed to present a formally correct work are present, the statistical implant is acceptable and properly detailed. The vastness of data and consistency of result are good hints at the soundness of the findings. 

 The conclusions are straightforward and apparently supported by evidence.   

Author Response

Reviewer 2

The authors present an interesting review on a topic that, is relatively new and poorly explored by now, albeit not exactly central in cardiovascaular medicine practice

 The description of the method is clear and concise, with sufficient details to understand the methods and the rationale.

 The language is correct, the paper is well written, every step needed to present a formally correct work are present, the statistical implant is acceptable and properly detailed. The vastness of data and consistency of result are good hints at the soundness of the findings.

 The conclusions are straightforward and apparently supported by evidence. 

Response: We are thankful for your positive comments, interest and time to review.